# Innovative data techniques for centrifugal pump optimization with machine learning and AI model

**Gaurav Sandeep Dave** [1]*, **Amar Pradeep Pandhare**[1], **Atul Prabhakar Kulkarni**[2], **Dhananjay Vasant Khankal**[3]

**1** Department of Mechanical Engineering, Sinhgad College of Engineering, Savitribai Phule Pune University, Pune, India, **2** Department of Mechanical Engineering, VIIT, Pune, India, **3** Department of Mechanical Engineering, Sinhgad College of Engineering, Savitribai Phule Pune University, Pune, India

* davegaurav.logic@gmail.com

## Abstract

In modern centrifugal pump machines (CPM), a data acquisition system encompassing software- hardware interfacing is essential for parameter recording. The quality of recorded data plays a crucial role and directly influences the data transformation phase in machine learning (ML) and deep learning (DL) models. The Dewesoft FFT DAQ system is designed to extract the high-quality data from the CPM based on sensor fusion technology. The data recorded from DAQ system undergoes thorough in-depth analysis, processing & transformation before being incorporated into machine learning (ML) or artificial intelligence models. This paper emphasizes the importance of data cleaning, pre-processing, and applying appropriate methodologies to transform raw data into a valuable resource that can be utilized by ML and AI models. Key techniques include Exploratory Data Analysis (EDA), Data Visualization, and Feature Engineering (FE), which collectively enhance data interpretability. Following these transformations, hypothesis testing validates the data's integrity, ensuring reliability for subsequent modeling. The validated data is employed to train machine learning classifiers and deep learning algorithms, targeting a 27.25% enhancement in operational efficiency based on F1 score. Additionally, it decreases model training time by 180 seconds, facilitating predictive maintenance of critical performance metrics and minimizing downtime. The assessment of model performance relies on Precision, Recall, and F1 score. This approach leverages recent advancements in data science to derive actionable insights from CPM data, facilitating more informed decision-making and optimization of pump operations.

## Introduction

Centrifugal Pump Machines are employed in a variety of applications because they are dependable, efficient, and versatile. However, because of the intricate architecture of the system and the volume of data acquired during operation, predicting

**Data availability statement:** The authors confirm that the data and codes supporting the findings of this study are available in the article and its supplementary materials. The corresponding author, G. S. Dave, agrees to share the data. The link for supporting data and codes from institutes HOD committee: https://github.com/hodmechscoe/research-data.

**Funding:** The author(s) received no specific funding for this work.

**Competing interests:** The authors have declared that no competing interests exist.

the performance and identifying faults in centrifugal pumps is an extremely difficult process. Machine learning and deep learning algorithms have demonstrated great potential in addressing these challenges by automating the process of data analysis and feature extraction. This study emphasizes the need for a theoretical framework connecting machine learning (ML) and artificial intelligence (AI) to predictive maintenance and anomaly detection in centrifugal pump monitoring [1]. Predictive maintenance employs condition monitoring principles, analyzing historical and real-time sensor data to foresee potential failures. Fault detection theories, including model-based and data-driven diagnosis frameworks, highlight the significance of identifying deviations from typical operational behavior. When appropriately integrated with these frameworks, ML and AI models can autonomously discern meaningful patterns from extensive sensor datasets, recognize early degradation signs, and initiate timely maintenance [2]. By aligning with established theories in condition monitoring and fault detection, this study enhances the empirical approach of the proposed methodology, ensuring it is rooted in reliable scientific principles, thus improving the reliability and interpretability of the predictive models created [3].

Recent studies have applied advanced signal processing and machine learning methods to detect fault of centrifugal pump. To identify the severity of cavitation, Azizi et al. [4] created a technique based on feature extraction for empirical mode decomposition (EMD). Kumar, Anil, et al. [5] suggested an enhanced deep convolutional neural network (CNN) utilizing acoustic imagery for detection of fault in the components of pump. Gao et al. [6] employed a hybrid feature selection method to enhance the precision of cavitation severity identification. Resende et al. [7] developed the TIP4.0 platform. for predictive maintenance (PdM) using CNN architecture with distributed edge computing. Pooja Vinayak Kamat et al. [8] employed K-means clustering and autoencoder- LSTM for anomaly trend analysis and classification. K-means clustering was used by Antonio L. Alfeo et al. [9] to improve the interpretation of model by achieving the best quality feature combination for a specific classification problem. Before modelling, exploratory data analysis (EDA) is usually carried out to identify patterns and informative aspects in the dataset. However, manual feature engineering by maintenance personnel has limitations. More automated feature learning approaches like semi-supervised learning are being explored.

Various studies have investigated methodologies such as support vector machines (SVMs) [10] for the prediction of cavitation and blockage of flow, variational mode decomposition for diagnosing faults in rolling bearings, and a data indicator-based deep learning network for the detection of multiple faults. Cyclostationary analysis, wavelet decomposition, and symmetric cross entropy of neutrosophic [2] sets have been utilized in the context of pump condition monitoring [8,9].

The procedure encompasses Exploratory Data Analysis (EDA), feature engineering, data transformation and validation, as well as machine learning and deep learning methodologies. EDA plays a crucial role in analyzing the characteristics of the dataset and identifying patterns and correlation between variables. Feature engineering is the method of creating or merging new dimensions from the unprocessed data to improve the quality of machine learning models. Data transformation,

standardization/normalization and validation are essential steps in data analysis process to ensure the reliability and consistency of data [11]. ML and AI techniques have been widely applied to centrifugal pump data for performance prediction and fault diagnosis.

Centrifugal pump machines are dependable and efficient; nonetheless, their complexity and vast operational data complicate performance forecasting and fault diagnostics. Advanced signal processing, machine learning (ML) and deep learning (DL) techniques tackle these challenges, although they possess limitations. Empirical mode decomposition (EMD) and feature extraction necessitate manual intervention, which is labour-intensive and susceptible to errors. Deep CNNs can identify flaws; however, they necessitate substantial computer resources and labelled data, which may not always be available. Hybrid methodologies such as k-means clustering or autoencoders enhance feature selection; yet, they may lack scalability and generalizability across diverse datasets. Numerous studies overlook multi-fault scenarios and automated feature learning techniques, such as semi-supervised learning, which constitutes a significant constraint to the research.

Machine learning methodologies are increasingly employed for pump fault detection but encounter issues related to nonlinearities, noise susceptibility, and generalizability. Recent investigations highlight the need for resilient models capable of precise fault diagnosis amid uncertainties. The efficacy of machine learning and deep learning models is significantly influenced by data handling quality [4]. A discernible void exists in the literature regarding data preprocessing and exploratory data analysis (EDA) techniques that proficiently scrutinize intricate patterns within multivariate sensor datasets [12]. This study aims to address this gap by proposing an enhanced preprocessing framework followed by hypothesis testing tailored for improved monitoring of centrifugal pumps.

This study seeks to create models by applying advanced data analysis and machine learning to centrifugal pump data, striving to create reliable models that can detect flow obstructions, forecast pump performance, identify multi faults, and enable predictive maintenance—ultimately improving the reliability and efficiency of centrifugal pump systems. Specifically, this research leverages the Dewesoft FFT DAQ system and integrates machine learning techniques to improve predictive accuracy and operational efficiency. The use of the Dewesoft FFT DAQ system provides high-resolution data acquisition capabilities, while the applied machine learning methods streamline the process of identifying faults and predicting performance with greater precision and less computational time [13]. This dual approach addresses existing limitations by automating feature extraction and enhancing scalability and reliability across diverse operational scenarios.

## Methodology

The data collected from the CPM has to be investigated and transformed before sending it to ML/DL Model [14]. The architecture flow of Exploratory Data Analysis (EDA) model should start from data analysis which helps to understand the statistics of CPM and its behavior [15–17]. It will help to understand whether the data recorded through DAQ has noise and outliers [18]. Data pre- processing is done after statistical analysis which helps to eliminate noise and outliers in data set [19]. After data pre-processing, data visualization becomes easy and helps to get insight of data [20,21]. Data visualization uses three approaches univariate, bi-variate and multivariate analysis to get insights of data [22]. Feature Engineering helps to eliminate the column and row feature with less information based on standard deviation and co-variance

Based on the requirement the standardization of data along with splitting the data set to training and testing is to be done based on ML/DL model [23–25]. Most important step after EDA is hypothesis testing to check whether the transformation done to raw data has make the data bias or not. The transformed data after passing hypothesis test should be ready to train for ML/ DL model [26–28].

### The basic architecture flow of the model

The experimental setup presented in the Fig 1 shows the hardware that includes the Centrifugal Pump Machine, Sensors & the data acquisition system. The experiment uses sensor fusion that extracts the data using Dewesoft FFT Analyser.

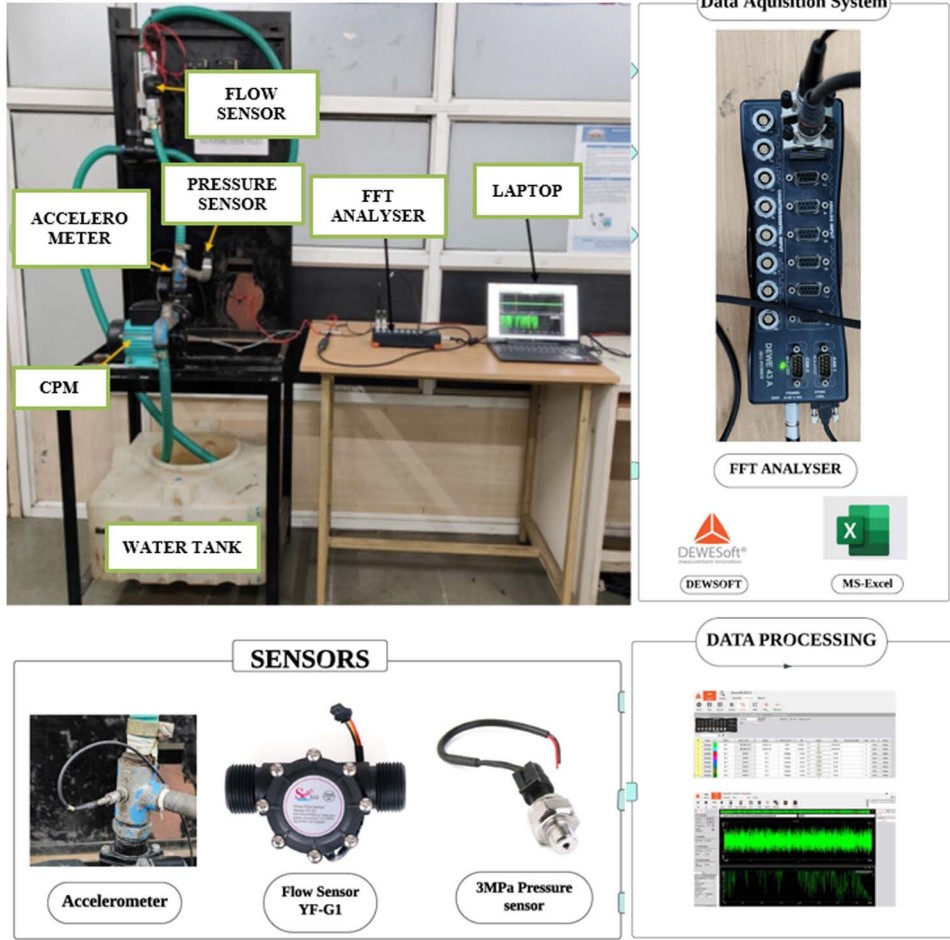

**Fig 1. The CPM with FFT DAQ System.**

The sensors include Accelerometer, Flow sensor and Pressure sensor along with the multimeter to record the current and voltage [13].

Dewesoft's FFT analyzer can perform real-time FFT analysis on unlimited input channels. It has a sampling rate of 200 kHz per channel, and a dynamic range of 160 dB in the time and frequency domains. The maximum frequency bandwidth that provides valid values without aliasing effects is the sample rate divided by 2.56. Fig 2 illustrates the fundamental design of the model. The specific & detailed subset of the model is illustrated in Fig 3, showcasing real-time exploratory data analysis (EDA) and feature engineering (FE). The extracted features are recorded in xlsx or csv files. The data extracted from the Centrifugal Pump Machine (CPM) via data acquisition system comprises 12 independent variables and 4 dependent variables, resulting in a dataset with 13 columns and 70,062 data points [13]. For further analysis, the data should be uploaded into a suitable environment such as Jupyter Notebook, Google Colab, or PyCharm

To get insight into the recorded data, exploratory data analysis (EDA) is utilized to to understand the underlying patterns, relationships, and insights within the dataset followed with modelling. As seen in Fig 2, the knowledge gained from EDA aids in the transformation of the data through the use of Feature Engineering (FE). The process of feature engineering can occasionally skew the data, leading to poor modelling. To prevent it, when transforming data, the hypothesis test should be used to determine whether data bias is possible. The converted data is prepared for training under the ML and

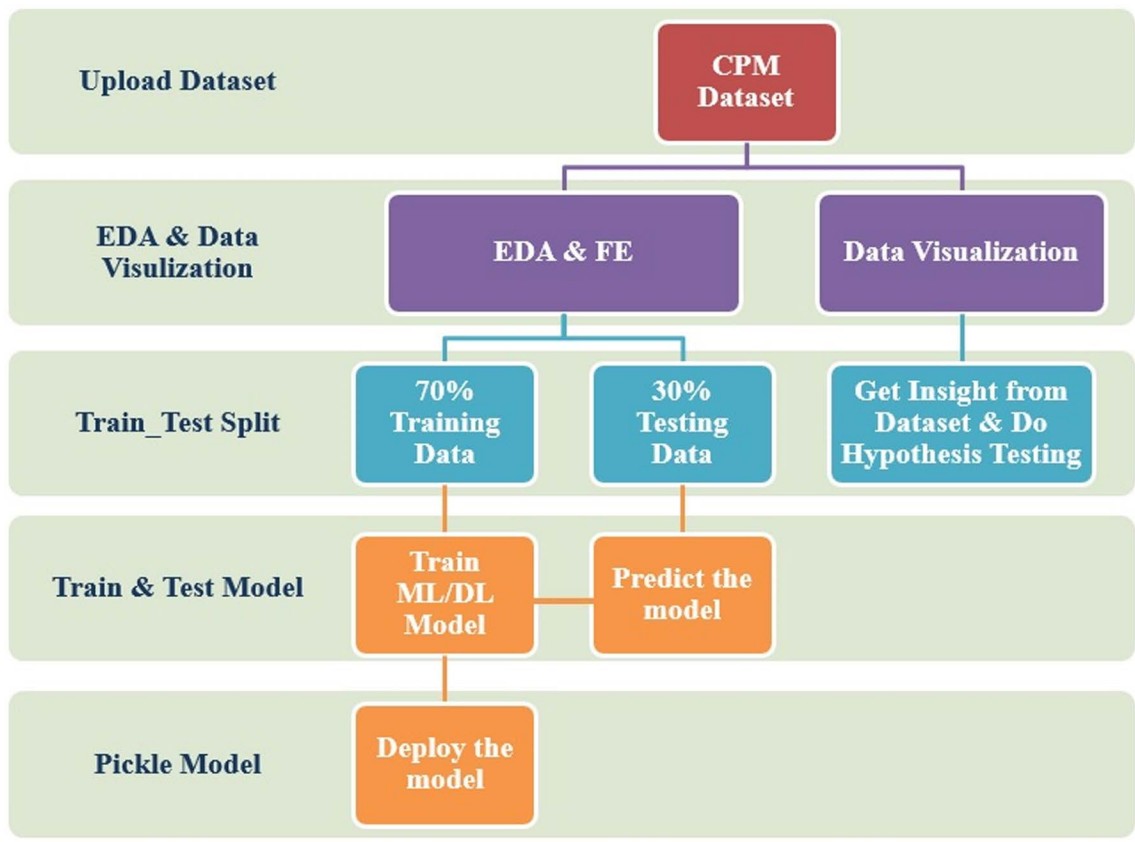

**Fig 2. The Basic architecture of the model.**

DL models if the hypothesis data validates the data set's quality. To proceed with production, the finest model is further pickled.

### Data analysis & pre-processing

Data is essential for machine learning models. High-quality input data enhances model performance. To increase the quality of data, Data preprocessing is to be performed which is consist of data analysis and data visualization. Real-time data is often contaminated with noise and outliers. Consequently, training models on raw data is ineffective. Data preprocessing transforms unrefined data into a structured format suitable for machine learning (ML) and deep learning (DL) models [29,30].

Primary steps for data cleaning consist of:

- Eliminating Null data

- Removing Duplicates in data

- Transforming Categorical data

- Dealing with noise & outliers

In various industries, including the field of manufacturing and mechanical engineering, the health

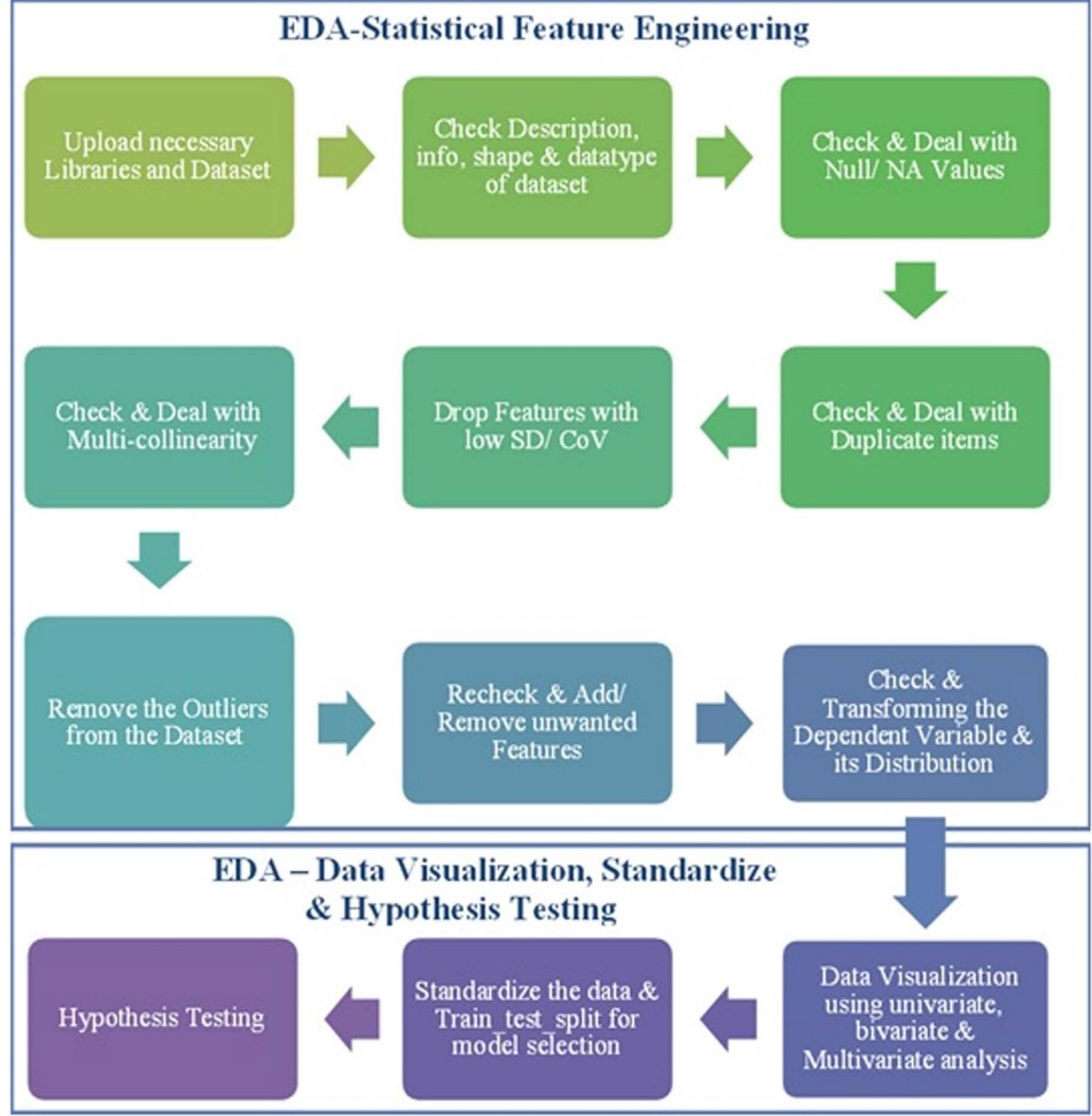

**Fig 3. EDA architecture for CPM Dataset.**

monitoring of equipment plays a crucial role in ensuring smooth operations and preventing unexpected failures. Among the many types of machinery used in these industries, centrifugal pumps are commonly employed for various applications, such as fluid transportation and circulation [31,32].

To effectively monitor the health of a CPM, it is important to collect and analyze relevant data from sensors and other monitoring devices. However, the collected data may often contain noise, outliers, or missing values, it may compromise the analysis's accuracy. As a result, data pre-processing methods address the issues and enhance data quality. One powerful approach for data pre-processing in the health monitoring system of a centrifugal pump is statistical & visualization. Statistical & visualization techniques allow analysts and engineers to gain insights into the data, detect anomalies, and

identify patterns. By visualizing the data, it becomes easier to understand its characteristics and make informed decisions regarding data cleaning and preprocessing steps which is important part of EDA [33]. EDA can be applied in various ways during the data pre-processing stage [34–36].

i. **Data cleaning:** Data Cleaning can help identify outliers or noisy data points that need to be removed or corrected. Histograms, box plots, and scatter plots are frequently used to show data distributions and spot abnormalities.

ii. **Missing data handling:** Statistics and visualization aids in understanding the patterns of missing data. By finding missing values, analysts can decide on appropriate strategies for imputation, such as mean or regression-based imputation, to fill in the gaps.

iii. **Feature engineering:** Feature Engineering helps in identifying relationships and correlations between different variables in the dataset. This knowledge can guide the selection of relevant features for further analysis and modelling.

iv. **Data transformation:** Data Transformation techniques can assist in exploring the distribution of variables and identifying skewness or non-normality. Such insights can guide data transformation steps, such as logarithmic or power transformations, to normalize the data for subsequent analyses.

Overall, incorporating statistical & visualization techniques into the data pre-processing stage of a HMS for a CPM enables a better understanding of the data and helps address challenges related to noise, outliers, missing values, and data transformation [37–39]. Consequently, this leads to higher operating efficiency, fewer downtime, better maintenance methods, and more precise and trustworthy health monitoring [39–41]. Recall that efficient data pre-processing is an essential component of any data analysis pipeline, and analysis is a useful tool in this process for the centrifugal pump's health monitoring system [42–44].

### Realtime exploratory data analysis on CPM – feature engineering

The Exploratory Data Analysis (EDA) & Feature Engineering (FE) includes methodologies such as statistical investigation, dealing with null & duplicate values, locating outliers, dealing with multicollinearity, re- checking, removing & adding unwanted features, check & transforming the dependent variable & its distribution [45–46]. The methodology as shown in Fig 3 also includes data visualization, discovering patterns, standardizing dataset & doing hypothesis testing to check the possibility of biasness before modelling.

After hypothesis testing the data is ready to train with different ML & DL models. The model generally works very well after EDA and FE as shown in Fig 3. Performing EDA is fundamental and comprises approximately 70% of the total effort in the analysis process. To begin the analysis of the data collected from the Centrifugal Pump Machine (CPM), various libraries are required, including Pandas, Numpy, Matplotlib.pyplot, Seaborn, Scipy, and sklearn (for preprocessing and train_test_split). Additionally, model_selection and imblearn libraries will be used. The first step involves using the Pandas library to read the data file. Following the data upload, the next step is to check the shape, information, five-point summary (Table 1), and data types of the dataset. The dataset contains 70,062 rows and 13 columns, with 7 float features, 4 integer features, 1 timestamp, and 1 object-categorical feature. The features Casing, Impeller, and Bearing provide vibration data, along with associated temperature values designated as C_Temp, I_Temp, and B_Temp. The DC_RA feature is documented to evaluate the roughness within the impeller casing, aimed at predicting cavitation. Additionally, the features Flow, Pressure, Current, and Voltage are recorded during operation to assist in identifying potential faults in the model. The Feature timestamp and condition are not displayed in the five-point summary table due to the data type not being integer or float.

The five-point summary provides a statistical overview, including the no. of observations, mean, standard deviation, minimum, 25th percentile, median, 75th percentile, and maximum values. As seen in Table 1, this summary aids in comprehending the central tendencies & distribution of data. The calculation of the standard deviation for quantiles (25%,

**Table 1. The 5-point summary of CPM dataset.**

|  | count | mean | std | min | 25% | 50% | 75% | max |
|---|---|---|---|---|---|---|---|---|
| CASING | 70062.0 | 387.419 | 193.5429 | 182.426 | 221.880 | 290.00 | 491.00 | 829.00 |
| C_TEMP\| | 70062.0 | 28.643 | 1.287 | 27.000 | 28.000 | 28.00 | 29.00 | 31.00 |
| IMPELLER | 70062.0 | 388.680 | 216.582 | 154.080 | 177.306 | 408.50 | 495.0 | 955.00 |
| I_TEMP | 70062.0 | 29.216 | 1.208 | 26.000 | 29.000 | 29.00 | 30.00 | 32.00 |
| BEARING | 70062.0 | 399.096 | 190.522 | 168.013 | 212.746 | 408.00 | 498.00 | 836.00 |
| B_TEMP | 70062.0 | 37.005 | 4.491 | 31.000 | 31.000 | 40.00 | 41.00 | 43.00 |
| FLOW | 70062.0 | 138.091 | 15.431 | 110.400 | 120.000 | 144.00 | 148.80 | 158.40 |
| PRESSURE | 70062.0 | 18.121 | 7.078 | 11.000 | 12.20 | 14.70 | 24.400 | 36.60 |
| DC_RA | 70062.0 | 3.773 | 0.906 | 2.340 | 2.340 | 4.332 | 4.332 | 4.466 |
| CURRENT | 70062.0 | 1.636 | 0.299 | 1.350 | 1.350 | 1.850 | 1.890 | 2.230 |
| VOLTAGE | 70062.0 | 181.819 | 56.549 | 142.000 | 142.000 | 142.00 | 234.00 | 291.00 |

50%, and 75%), the sample population, and the mean for the sample population are used for the data obtained. So, in essence, the percentiles split an ordered sample into 100 equal parts and return the corresponding data values [47–49]. To ensure data integrity and quality, the first step involves checking and handling any Null or NA values in the dataset. These missing values should be treated and imputed using methodologies such as replacing them with the mean, median, or other appropriate functions. However, in the case of the CPM dataset under consideration, there are no Null or NA values present, as confirmed by reference. Next, it is essential to address any potential duplicate items within the dataset. This process involves implementing functions that can identify and remove rows or features with identical values across all rows, effectively eliminating any duplicates. Fortunately, the CPM dataset also does not contain any duplicate items, as indicated by the same reference [50].

This ensures that the data is clean and reliable for further analysis. When analyzing data, it is often necessary to drop features with low standard deviation (SD) or coefficient of variance (CoV) to avoid the curse of dimensionality. The CoV is a statistical measure of relative variability, where a lower CoV indicates less dispersion around the mean. As CoV is directly dependent on standard deviation, it provides a normalized measure of dispersion within the data distribution. In practice, CoVs below 1 are considered low variance, while CoVs above 1are regarded as high variance. Essentially, SD measures the raw variability, whereas CoV indicates variability relative to the mean, facilitating comparison between variables using a standardized dispersion ratio [51]. The step involves dropping features with low SD or CoV, considering their limited contribution to the data's variability. Although CoV can identify and eliminate more features than SD, it is highly sensitive to outliers, making it crucial to use subject expertise to make the final decision. While SD is more popular, CoV is more effective when comparing different features within a dataset, provided there are no outliers and the feature units are the same. Setting a threshold of 0.2, the function eliminates features with values below this threshold. However, upon applying this approach, it was observed that every feature in the dataset had an SD above 0.20, so no features were dropped. This reinforces the recommendation to use the SD approach in most cases due to its robustness against outliers and its general applicability across various datasets [52]. In a centrifugal pump, removing unwanted low variance features can improve performance and efficiency. The higher variance in sensor data from the mean indicate potential faults.

SD-based feature extraction retains only high variance features, enhancing effectiveness. In contrast, PCA may reorganize data but fails to eliminate inefficiencies. Therefore, akin to a filter in a pump improving flow efficiency, SD filtering in data analysis preserves relevant data while discarding features with information below the 0.2 SD threshold.

A number of preparation procedures were carried out to guarantee the dataset's efficacy and integrity for machine learning (ML) and deep learning (DL) models. Multicollinearity, a situation where features contain redundant information, was addressed by assessing the correlation between features and setting a threshold of 70%. Features exceeding this

threshold were identified for removal, but a subject matter expert was consulted to ensure essential features were not discarded; three important features initially dropped due to multicollinearity were retrieved [53]. Outliers were identified and eliminated using the Scipy-stats library, targeting values greater than the 75th percentile and less than the 25th percentile. Rows with 90% zero values were also removed, significantly improving model performance [54]. Analysis revealed an imbalance in the dependent variables, with the GHC dependent variable being predominant. The Synthetic Minority Over-Sampling Technique (SMOTE) from imblearn library was used to increase minority class samples and achieve balance. Additionally, categorical dependent variables were converted to numerical categories or one-hot encoded for DL models. Combining these preprocessing steps ensured the data was well-prepared, enhancing the reliability and accuracy of subsequent analyses and models [55].

The CPM dataset is further processed to 70% training and 30% testing. The cleaned and processed data is split to train-test using sklearn.model selection library. Standardization of data is transformation process to convert data into more usable and sensible information. Sklearn. preprocessing-Standard Scaler library is used after splitting data set when the features have units with large SD and Mean. After Data Preprocessing using statistical and visualization approach it is important to do hypothesis testing to find out whether the data got biased during the Feature Engineering procedure.

## Hypothesis testing

Z-Test and T-test both uses average of sample data to calculate the probability as shown in Fig 4. The Z-test is used when the population standard deviation is known, or the sample size is large (n ≥ 30). The T-test is used when the population standard deviation is unknown, and the sample size is small (n <30). So, the possibility of T-test is eliminated. The Anova test is used when the analysis of variance is to be done. It uses one way and two-way approach to do it. If the population standard deviation is known, use the Z-test regardless of the sample size. If it is unknown and the sample is small, use the T-test. After deciding the test based on sample population Null Hypothesis and Alternate Hypothesis should be decided.

The p-value must be calculated for the sample data, which will inform the outcomes of the null and alternative hypothesis tests for both the z-test and ANOVA test. Therefore, the hypothesis will be applied for statistical comparison of CPM data before and after preprocessing.

The Z-test failed to reject the null hypothesis, indicating no significant difference between the means of the training and test sets for the "casing" feature. Likewise, the ANOVA test failed to reject the null hypothesis, suggesting that the means of the original, training, and test sets are statistically equal. The findings of the Z-test and ANOVA test suggest that no bias occurred during feature engineering, and the data is suitable for modelling.

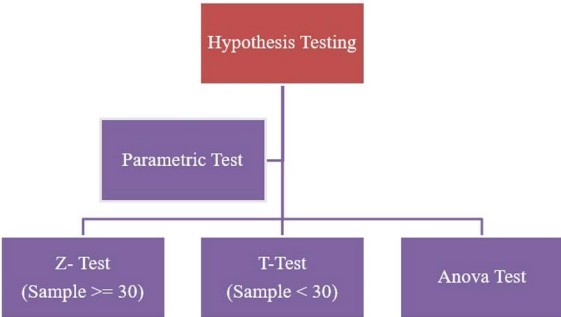

**Fig 4. Hypothesis testing of CPM dataset.**

## Results & discussion

After data cleaning and processing through Exploratory Data Analysis (EDA) and statistical feature engineering, the final stages involve data visualization, train-test splitting, standardization, and hypothesis testing. Merging EDA with data visualization accelerates model understanding and deployment. Data visualization is a crucial analysis technique for discovering patterns and visualizing statistical data, and it is divided into three main categories. Univariate analysis, which examines one variable at a time, has been employed to investigate 03 different graphs for the same variables and understand the distribution and characteristics of individual variables in the CPM data [56,57].

### Box plot analysis

The box plots [58] for "Casing," "Bearing," and "Impeller" reveal similar distributions of vibration, with medians around 300 to 400 G and interquartile ranges spanning from approximately 200 to 500 as shown in Fig 5. The whiskers indicate the overall spread of the vibration data, extending from about 200 to 800 G for each variable, with no significant outliers present. Understanding the fundamental patterns and distributions in the dataset is aided by this analysis, which offers a clear picture of each variable's central tendency, dispersion, and range of values

### Pie chart analysis

As observed in pie chart [59] for dependent variables – Good Health Condition (GHC), Impeller Fault (IF), Impeller & Bearing Fault (IBF) and Misalignment (MA) is distributed with a ratio of 30.5%, 27.1%, 27.1% & 15.3% respectively as shown in Fig 6. The data is slightly misbalanced within GHC, IF, IBF and MA.

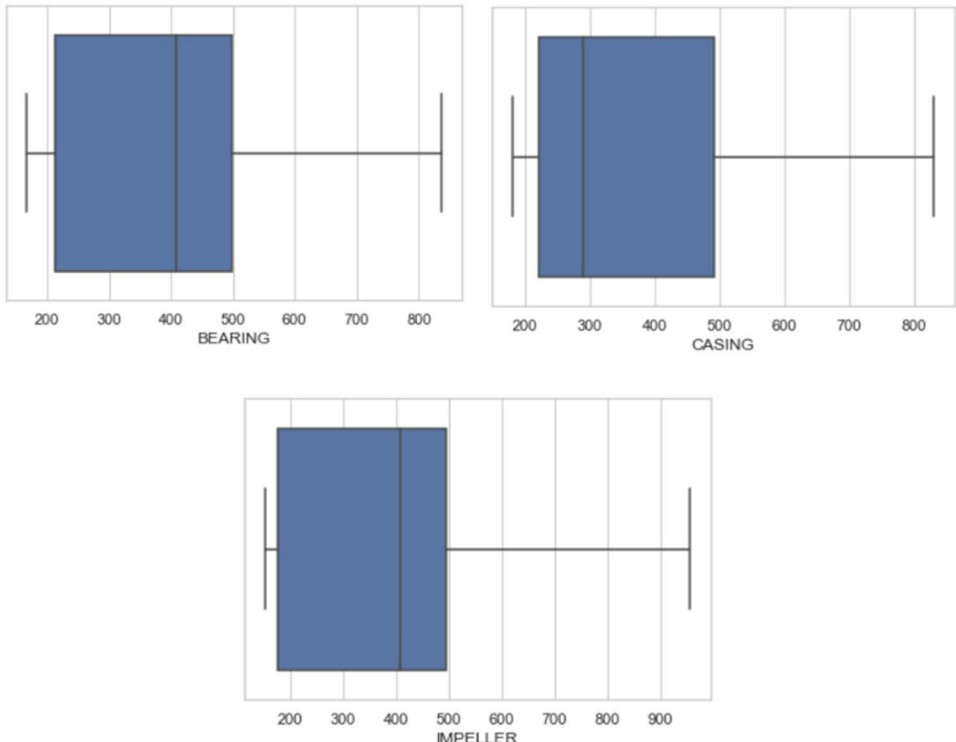

**Fig 5. Boxplot analysis of casing, impeller & bearing vibration.**

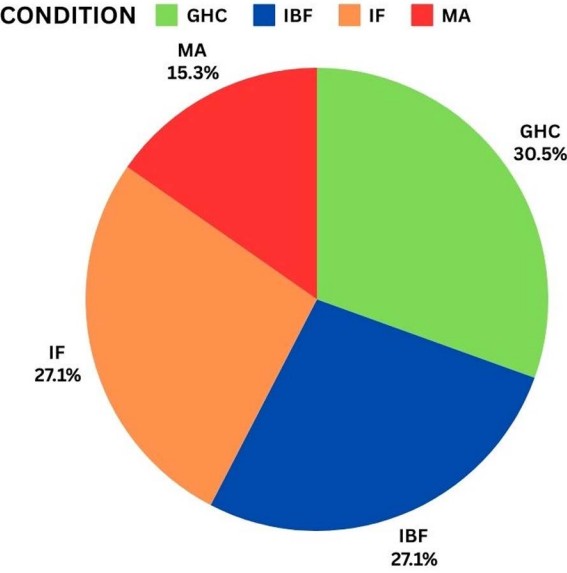

Fig 6. Pie chart of dependent variables.

## Line plot analysis

The line plots [60] display the variations in pressure over a specified range for three components: casing, impeller, and bearing as shown in Fig 7. Each figure displays an undefined measure on x-axis and pressure on y-axis, likely related to time or operational cycles. The casing feature varies between the vibration range of 200-800 G with pressure varying from 12.5 -29 KPa. The impeller feature varies between the vibration range of 200-950 G with the pressure range of 10-36 KPa. The bearing feature varies between the vibration range of 200-850 G with the pressure range of 11-36 KPa. All three components exhibit periods of stability and fluctuation. The recurring patterns around specific marks (200, 400, 500) suggest these could be critical operational phases or maintenance cycles. Spikes and sharp drops in pressure are areas of interest for further investigation to understand the underlying causes, such as equipment wear, load changes, or procedural adjustments. The plots show that while there are fluctuations, there are also significant periods where the pressure is relatively stable, indicating normal operational conditions. These insights can guide maintenance schedules, operational adjustments, and additional research to improve the system's dependability and effectiveness.

## Bar plot analysis

The bar plot [61] displays the variations in pressure, current, and voltage across four conditions: GHC (Good Health Condition), IF (Impeller Failure), IBF (Impeller & Bearing Failure), and MA (Misalignment) as shown in Fig 8 For pressure, GHC shows the highest value around 21 KPa, while IF and IBF have lower values near 15 KPa, and MA is slightly higher at 17 KPa. In terms of current, GHC and MA exhibit similar values around 1.6 A, IF has the lowest at 1.3 A, and IBF shows the highest at 1.8 A. For voltage, GHC is about 175 V, IF and IBF are around 125 V, and MA has the highest at 200 V. These plots indicate the health and maintenance status of the system, with GHC showing optimal conditions, IF and IBF indicating failures, and MA suggesting the need for maintenance.

## The co-relation heatmap analysis

The heatmap [62] shown in Fig 9 displays the correlation matrix for various features including casing, impeller, bearing, temperatures (C_TEMP, I_TEMP, B_TEMP), flow, pressure, DC_RA, current, and voltage. Dark hues on the color scale indicate a

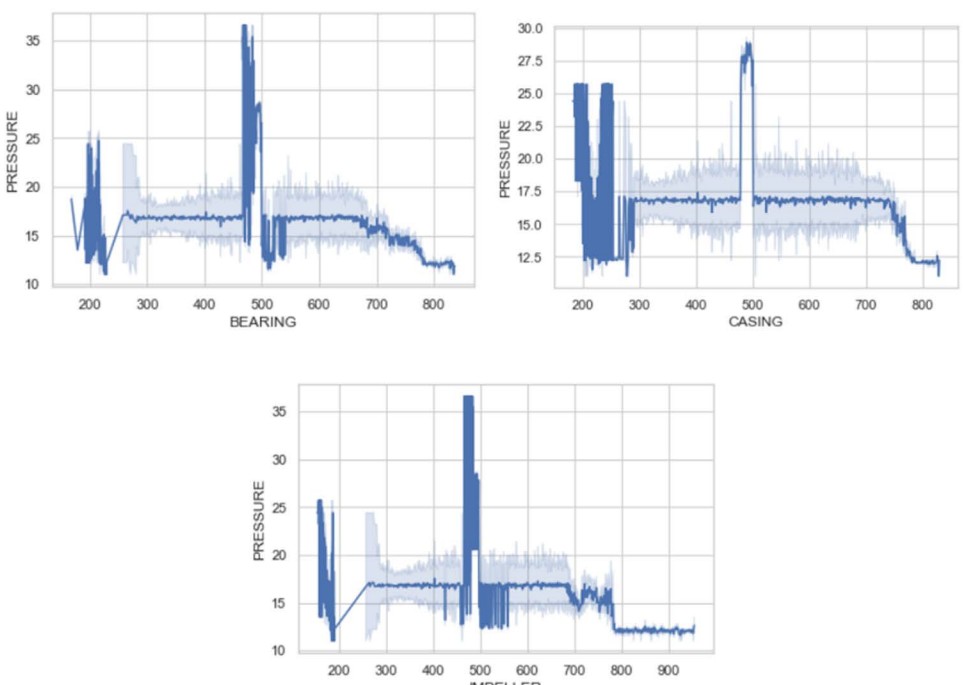

**Fig 7. Bivariate analysis using line plot for casing, bearing & impeller vs pressure.**

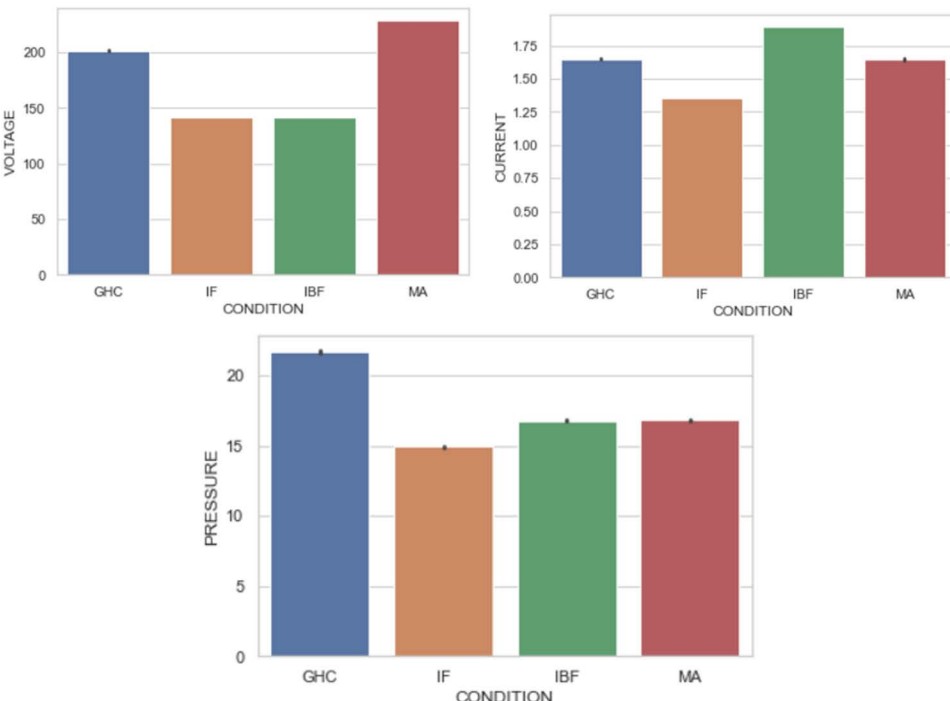

**Fig 8. Bivariate analysis using bar plot for pressure, current & voltage vs condition.**

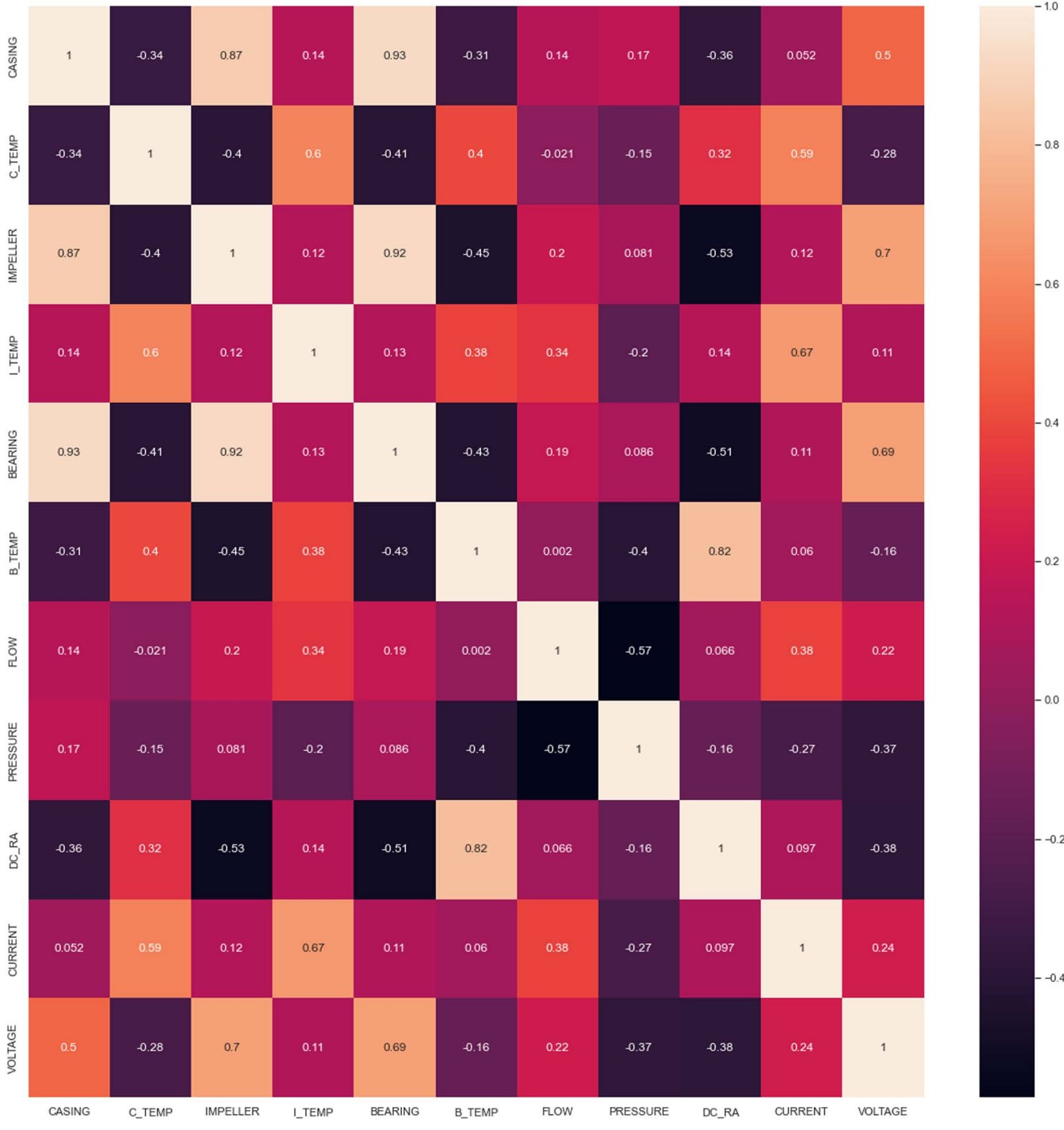

**Fig 9. Heatmap of CPM dataset.**

negative correlation, whereas light hues indicate a positive correlation. This heatmap is essential for identifying which variables are interrelated and which are independent, aiding in predictive modeling and operational decisions. The heatmap illustrates the significance of features and the relationship between variables. The heat maps demonstrate a significant relationship between DC_RA and B_temp, indicating that increased cavitation is associated with higher bearing temperatures. Bearing temperature is indicative of cavitation in both the casing of impeller and impeller. In a Good Health Condition (GHC), there is a positive relationship between flow rate and pressure. Conversely, 75% of the data reflects defective conditions, where MA greatly diminishes flow. As a result, the heat map illustrates a negative correlation that detrimentally affects pump performance. A positive correlation of 0.70 is seen between impeller vibration and voltage, indicating that voltage increases when the impeller is damaged. The correlation exceeding 0.70 between the variables is instrumental in predicting maintenance faults.

## Scatter clustering analysis

The link between impeller readings (x-axis) and casing readings (y-axis) is depicted in a scatter plot [63] in Fig 10, where bearing values are represented by color intensity. For instance, data points with high impeller readings (around 700-900

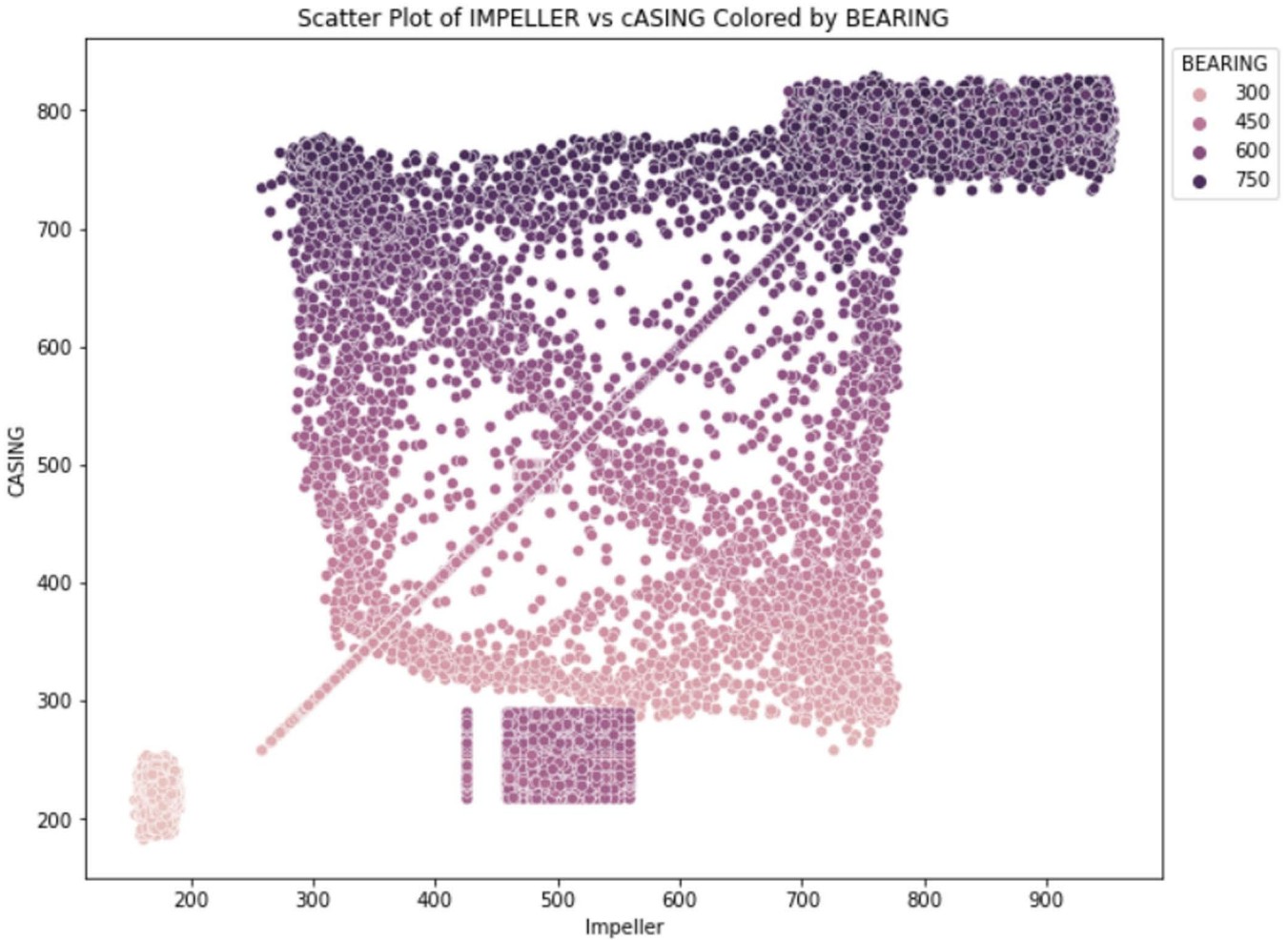

**Fig 10. Impeller vs casing with bearing scatter.**

G) and high casing readings (around 500- 700 G) are marked with darker colors, indicating higher bearing values. This plot helps identify clusters and outliers, providing insights into typical and atypical operational conditions. For example, a dense cluster of points around impeller readings of 300-500 G and casing readings of 400- 600 G with moderate bearing values suggests a common operational range, while isolated points might indicate anomalies. The existence of data clusters about 750G may signify malfunction in the CPM system, suggesting the necessity for maintenance by the support staff. Faults exceeding 750G signify shaft misalignment, while those below 300G may suggest bearing or impeller defects.

**K-means clustering analysis**

Fig 11 presents a 3D scatter plot [64] showcasing the relationships between casing, casing temperature (C_TEMP), and bearing values. The data points are plotted in three-dimensional space, providing a more comprehensive view of how these variables interact. For example, clusters of points around a casing value of 500G, C_TEMP of 150, and bearing value of 600G suggest stable operating conditions. Outliers in the plot, such as points far removed from these clusters, indicate potential issues or extreme operational conditions. This 3D visualization is particularly useful for detecting multi-variable interactions and anomalies that may not be visible in two-dimensional plots.

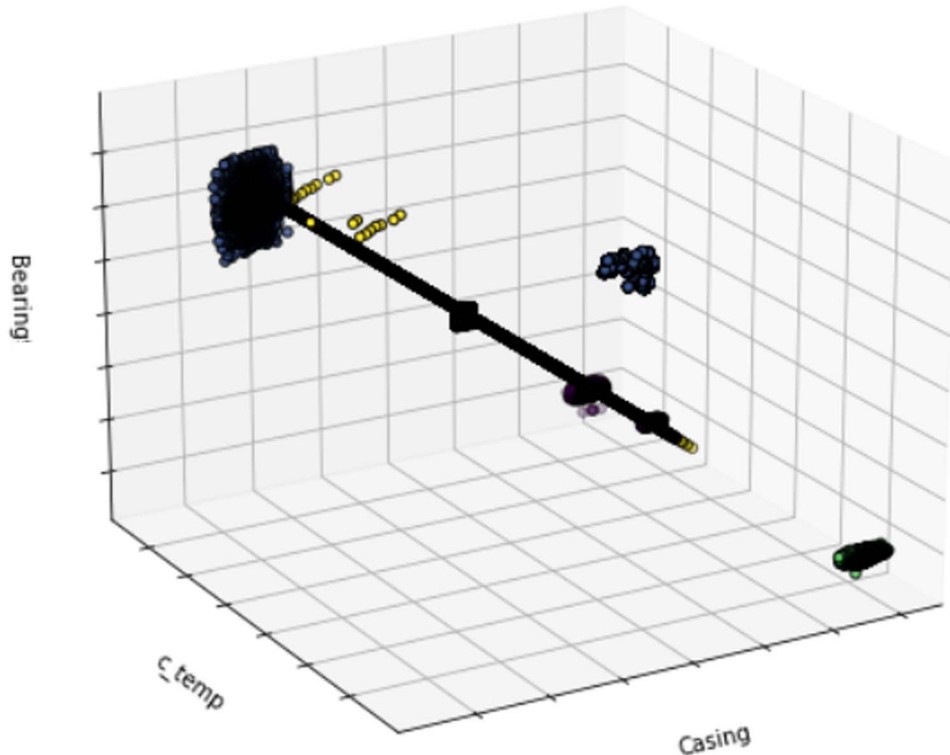

**Fig 11. K-means clustering of bearing vs c_temp vs casing.**

Through a comprehensive examination of the aforementioned graphical representations delineated in Figs 10 and 11, one can extract significant insights pertaining to the operational dynamics inherent within the system, ascertain critical correlations, and elucidate specific domains that necessitate additional scrutiny to guarantee the reliability and performance of the system.

For instance, consistent pressure values around 15-25 KPa in conjunction with moderate impeller and bearing values indicate stable operations, while deviations from these ranges could signal potential issues. The integration of heatmaps, scatter clustering analysis, and K-Means clustering analysis enables maintenance personnel or supervisors to statistically assess and visually identify potential faults, hence minimizing industrial downtime.

**ML classifier models**

Logistic Regression [65] as seen in Fig 12 represents a widely utilized supervised learning methodology that addresses both binary and multi-class classification challenges. This technique characterizes the likelihood of a data point being

| LOGISTIC REGRESSION CLASSIFIER | | | | | |
|---|---|---|---|---|---|
| TARGET / OUTPUT | GHC | IF | IBF | MA | SUM |
| GHC | 5266<br>17.55% | 0<br>0.00% | 0<br>0.00% | 2235<br>7.45% | 7501<br>70.20%<br>29.80% |
| IF | 0<br>0.00% | 6631<br>22.10% | 904<br>3.01% | 0<br>0.00% | 7535<br>88.00%<br>12.00% |
| IBF | 0<br>0.00% | 0<br>0.00% | 5212<br>17.37% | 2342<br>7.81% | 7554<br>69.00%<br>31.00% |
| MA | 0<br>0.00% | 0<br>0.00% | 3115<br>10.38% | 4301<br>14.33% | 7416<br>58.00%<br>42.00% |
| SUM | 5266<br>100.00%<br>0.00% | 6631<br>100.00%<br>0.00% | 9231<br>56.46%<br>43.54% | 8878<br>48.45%<br>51.55% | 21410 / 30006<br>71.35%<br>28.65% |

**Fig 12. Logistic Classifier Confusion Matrix.**

associated with a specific class by employing the logistic (sigmoid) function. In contrast to linear regression, which forecasts continuous variables, logistic regression is designed to estimate categorical outcomes and is highly esteemed for its straightforwardness, efficiency, and interpretability within practical applications.

The logistic regression classifier exhibits a moderate level of overall efficacy, achieving an accuracy rate of 71.35% and a corresponding misclassification rate of 28.65%. The model demonstrates a high degree of precision in predicting the IF class (88% correct), followed by the GHC class (70.2%) and the IBF class (69%), while it encounters considerable challenges with the MA class (58%). Instances of misclassification are predominantly observed between the MA and IBF classes, indicating a potential overlap or ambiguity within the feature space pertaining to these categories. In summary, although the model performs adequately with respect to two classes (GHC and IF), there remains a necessity for enhancements to facilitate improved differentiation between IBF and MA predictions.

The data in Table 2 indicates that the IF class exhibits the highest precision at 88%, with GHC, IBF, and MA following at 70.2%, 69%, and 58%, respectively. Both GHC and IF achieve perfect recall (1.0), whereas IBF and MA demonstrate significantly lower values at 56.46% and 48.45%, respectively, suggesting difficulties in accurately identifying relevant instances. The F1-scores are highest for IF (93.62%) and GHC (82.49%), with IBF (62.10%) and MA (52.79%) reflecting poorer performance. The Macro-F1 score of 72.75% indicates a reasonable yet inconsistent performance across different classes, while the Weighted-F1 score of 69.89% highlights the effect of class distribution on the model's efficacy. In conclusion, the model performs commendably in certain classes, particularly IF, but faces challenges with IBF and MA, signifying the need for further optimization, feature enhancement, or model refinement to enhance overall dependability.

The Naive Bayes Classifier as shown in Fig 13 is a straightforward, quick, and probabilistic machine learning algorithm. It determines the likelihood that a data point belongs to a class and makes the "naive" assumption that features are independent of one another. It's commonly used for multiclass classification problems. Despite its simplicity, it works well on large datasets and high-dimensional data, though its performance can drop when features are highly correlated. Among the most popular machine learning classifiers for the study include naïve bayes, decision trees, SVM, KNN, and random forests [66–71].

The Gaussian Naïve Bayes classifier's confusion matrix (Fig 13) shows how well it performs in the four classes of misalignment (MA), impeller fault (IF), impeller bearing fault (IBF), and good health condition (GHC). It shows that MA class is perfectly classified with no misclassifications. GHC class has 7428 correct classifications and 73 misclassified with MA. IBF has 7538 correct classifications and 8 misclassified as MA too.

This Naïve Bayes Classifier indicates high overall accuracy with minor misclassifications. The algorithm has 99.73% accuracy followed with recall. As observed in Table 3 the algorithm has misclassified 73 observations for GHC and 8 observations with total misclassification rate of 0.3%.

The Support Vector Classifier (SVC) [71–73] as shown in Fig 14 functions as a supervised learning algorithm, designed to categorize data into distinct classes. It achieves this by identifying the maximum range between margin of various class

**Table 2. Statistic Result of *Logistic Regression Classifier*.**

| Class Name | Precision | 1-Precision | Recall | 1-Recall | f1-score |
|---|---|---|---|---|---|
| GHC | 0.7020 | 0.2980 | 1.0000 | 0.0000 | 0.8249 |
| IF | 0.8800 | 0.1200 | 1.0000 | 0.0000 | 0.9362 |
| IBF | 0.6900 | 0.3100 | 0.5646 | 0.4354 | 0.6210 |
| MA | 0.5800 | 0.4200 | 0.4845 | 0.5155 | 0.5279 |
| Accuracy | 0.7135 | | | | |
| Misclassification Rate | 0.2865 | | | | |
| Macro-F1 | 0.7275 | | | | |
| Weighted-F1 | 0.6989 | | | | |

| Naïve Bayes Classifier | | | | | |
|---|---|---|---|---|---|
| TARGET / OUTPUT | GHC | IF | IBF | MA | SUM |
| GHC | 7428 24.74% | 0 0.00% | 0 0.00% | 73 0.24% | 7501 99.03% 0.97% |
| IF | 0 0.00% | 7535 25.10% | 22 0.07% | 0 0.00% | 7557 99.71% 0.29% |
| IBF | 0 0.00% | 0 0.00% | 7538 25.11% | 8 0.03% | 7546 99.89% 0.11% |
| MA | 0 0.00% | 0 0.00% | 0 0.00% | 7416 24.70% | 7416 100.00% 0.00% |
| SUM | 7428 100.00% 0.00% | 7535 100.00% 0.00% | 7560 99.71% 0.29% | 7497 98.92% 1.08% | 29917 / 30020 99.66% 0.34% |

**Fig 13. Naïve Bayes Classifier Confusion Matrix.**

**Table 3. Statistic Result of *Naïve Bayes Classifier.***

| Class Name | Precision | 1-Precision | Recall | 1-Recall | f1-score |
|---|---|---|---|---|---|
| GHC | 0.990 | 0.010 | 1.000 | 0.000 | 0.995 |
| IF | 0.997 | 0.003 | 1.000 | 0.000 | 0.999 |
| IBF | 0.999 | 0.001 | 0.997 | 0.003 | 0.998 |
| MA | 1.000 | 0.000 | 0.989 | 0.011 | 0.995 |
| Accuracy | 0.997 | | | | |
| Misclassification Rate | 0.003 | | | | |
| Macro-F1 | 0.997 | | | | |
| Weighted-F1 | 0.997 | | | | |

labels and plotting the optimal hyperplane accordingly. The method demonstrates strong performance in high-dimensional spaces and is proficient in handling both linear and non-linear classes through the hyperparameter of kernel functions. The Support Vector Classifier (SVC) demonstrates resilience to overfitting, particularly in scenarios where the dimensionality surpasses the sample size. However, it may incur significant computational costs when applied to large datasets.

The SVC confusion matrix (Fig 14) shows the classifier's ability to correctly classify instances across four classes. GHC and MA have perfect classification with all instances correctly identified. Class IF has 7413 correct classifications with 22 instances misclassified into Class IBF. Class IBF has 7227 correct classifications with 311 instances misclassified into Class IF.

This suggests the accuracy been dropped with some confusion between Classes IF and IBF. As observed in the Table 4 the Recall for SVC is 98.89% and F1scores is 97.80%. The SVC Classifier has done good in detecting GHC and MA.

| SUPPORT VECTOR CLASSIFIER | | | | | |
|---|---|---|---|---|---|
| TARGET / OUTPUT | GHC | IF | IBF | MA | SUM |
| GHC | 7501 / 25.10% | 0 / 0.00% | 0 / 0.00% | 0 / 0.00% | 7501 / 100.00% / 0.00% |
| IF | 0 / 0.00% | 7413 / 24.80% | 22 / 0.07% | 0 / 0.00% | 7435 / 99.70% / 0.30% |
| IBF | 0 / 0.00% | 311 / 1.04% | 7227 / 24.18% | 0 / 0.00% | 7538 / 95.87% / 4.13% |
| MA | 0 / 0.00% | 0 / 0.00% | 0 / 0.00% | 7416 / 24.81% | 7416 / 100.00% / 0.00% |
| SUM | 7501 / 100.00% / 0.00% | 7724 / 95.97% / 4.03% | 7249 / 99.70% / 0.30% | 7416 / 100.00% / 0.00% | 29557 / 29890 / 98.89% / 1.11% |

**Fig 14. Confusion Matrix for Support Vector Classifier.**

**Table 4. Statistic Result of Support Vector Classifier.**

| Class Name | Precision | 1-Precision | Recall | 1-Recall | f1-score |
|---|---|---|---|---|---|
| GHC | 1.0000 | 0.0000 | 1.0000 | 0.0000 | 1.0000 |
| IF | 0.9970 | 0.0030 | 0.9597 | 0.0403 | 0.9780 |
| IBF | 0.9587 | 0.0413 | 0.9970 | 0.0030 | 0.9775 |
| MA | 1.0000 | 0.0000 | 1.0000 | 0.0000 | 1.0000 |
| Accuracy | 0.9889 | | | | |
| Misclassification Rate | 0.0111 | | | | |
| Macro-F1 | 0.9889 | | | | |
| Weighted-F1 | 0.9889 | | | | |

While major misclassification is done for IF & IB with 22 and 311 observations as misclassification. The SVC models is not as effective as Navie Bayes Classifier.

The Classifier - Random Forest as shown in Fig 15 functions as an ensemble learning technique, constructing numerous decision trees throughout the training process. It integrates the predictions of these trees to enhance accuracy and mitigate the risk of overfitting [74,75]. The trees are constructed using a random subset of the data and features, enhancing the model's robustness and its resistance to noise. Random Forest demonstrates strong performance across various classification tasks and efficiently manages large datasets and high-dimensional spaces; however, training can be slower when utilizing a substantial number of trees.

The Random Forest classifier's confusion matrix reveals near-perfect performance. Class GHC has 7500 correct classifications with just 1 misclassification as MA. Class IF and Class MA are perfectly classified with all instances correctly identified. Class IBF has 7546 correct classifications with 8 misclassifications. The decision tree diagram visualizes the hierarchical structure and decision rules used by the Random Forest model, illustrating the paths taken to reach classifications. The Random Forest with CPM data set has 99.99% accuracy. The Fig 14 shows that Random Forest ensemble technique has only 1 misclassified observation, i.e., in GHC. The model demonstrates superior performance compared to others; however, in relation to Figs 13 and 14, the data exhibits complexity. It is established that complex data can lead to overfitting in random forest models. Therefore, complete reliance on the random forest model for complex datasets is not recommended.

The F1 Score recorded is 99.99% while Recall is 99.98% as shown in Table 5. Overall, these figures highlight the effectiveness of each model, with Random Forest showing the highest accuracy, followed by Naïve Bayes and then SVC. The decision tree for Random Forest provides insight into the model's decision-making process.

The Table 6 displays the comparative performance assessment of three machine learning models—Logistic Regression Classifier, Naïve Bayes Classifier, Support Vector Classifier, and Random Forest Classifier—according to three principal performance metrics: Precision, Recall, and F1-Score with respect to computational time. These criteria were selected to offer a thorough evaluation of the models' proficiency in executing categorization tasks.

1. **Logistic Regression Classifier:** The Logistic Regression Classifier demonstrates a Precision of 76.23%, Recall of 71.30%, and an F1-Score of 72.75%. While its performance is modest compared to advanced models, it provides a solid baseline and efficient training duration. Its lower Precision, Recall, and F1-Score signify weaker predictive performance compared to other evaluated machine learning models. It is particularly effective for linearly separable datasets and scenarios requiring interpretability. The computation time of 150 seconds makes it more efficient than more complex models.

2. **Naïve Bayes Classifier**: The Naïve Bayes Classifier attains a Precision of 99.73%, Recall of 99.02%, and an F1-Score of 99.51%, demonstrating its efficacy in accurately classifying pertinent cases while sustaining a balance between Precision and Recall. Its performance indicates efficacy in situations with probabilistic interdependencies among characteristics. The time taken for computation is 180 seconds which is lowest of all ML models.

| Random Forest Classifier | | | | | |
|---|---|---|---|---|---|
| TARGET / OUTPUT | GHC | IF | IBF | MA | SUM |
| GHC | 7500 / 25.00% | 0 / 0.00% | 0 / 0.00% | 1 / 0.00% | 7501 / 99.99% / 0.01% |
| IF | 0 / 0.00% | 7535 / 25.11% | 0 / 0.00% | 0 / 0.00% | 7535 / 100.00% / 0.00% |
| IBF | 0 / 0.00% | 0 / 0.00% | 7546 / 25.15% | 8 / 0.03% | 7554 / 99.89% / 0.11% |
| MA | 0 / 0.00% | 0 / 0.00% | 0 / 0.00% | 7416 / 24.72% | 7416 / 100.00% / 0.00% |
| SUM | 7500 / 100.00% / 0.00% | 7535 / 100.00% / 0.00% | 7546 / 100.00% / 0.00% | 7425 / 99.88% / 0.12% | 29997 / 30006 / 99.97% / 0.03% |

**Fig 15. RF Tree and Confusion Matrix for Random Forest.**

**Table 5. Statistic Result of Random Forest Classifier.**

| Class Name | Precision | 1-Precision | Recall | 1-Recall | f1-score |
|---|---|---|---|---|---|
| GHC | 1.000 | 0.000 | 1.000 | 0.000 | 0.999 |
| IF | 1.000 | 0.000 | 1.000 | 0.000 | 1.000 |
| IBF | 0.989 | 0.001 | 0.998 | 0.000 | 0.999 |
| MA | 1.000 | 0.000 | 1.000 | 0.001 | 1.000 |
| Accuracy | 9.999 | | | | |
| Misclassification Rate | 0.001 | | | | |
| Macro-F1 | 9.999 | | | | |
| Weighted-F1 | 9.999 | | | | |

**Table 6. Performance Table for ML Models.**

| Sr. No. | Type of ML Model | Percentage Performance Parameter | | | |
|---|---|---|---|---|---|
| | | Precision | Recall | F1-Score | Computation Time (Seconds) |
| 1. Logistic Regression Classifier | | 76.23 | 71.30 | 72.75 | 150 |
| 2. Naïve Bayes Classifier | | 99.73 | 99.02 | 99.51 | 180 |
| 3. Support Vector Classifier | | 98.84 | 98.89 | 97.80 | 300 |
| 4. Random Forest Classifier | | 99.99 | 99.98 | 99.99 | 360 |

3. **Support Vector Classifier**: Exhibits marginally reduced Precision (98.84%), Recall (98.89%), and F1-Score (97.80%) relative to alternative models. Although effective, its comparatively lower F1-Score suggests that further fine-tuning or feature engineering may be necessary to improve its performance in intricate data situations. The time taken to train the SVC model is 300 seconds.

4. **Random Forest Classifier**: Surpasses the previous models with nearly flawless Precision (99.99%), Recall (99.98%), and F1-Score (99.99%). These results highlight its strength and proficiency in managing varied datasets, perhaps attributable to its ensemble learning methodology and adept management of feature interactions and noise. The time taken to train the RFC model is highest 360 seconds as compared to other ML models.

The Fig 16 highlights that the Naïve Bayes Classifier is the most suitable model for the given dataset, offering superior overall performance for complex dataset with optimal computational time. The findings demonstrate the importance of model selection in achieving high reliability and accuracy for classification tasks in industrial applications.

**AI deep learning model**

An Artificial Neural Network (ANN) was utilized for classification via deep learning. The architecture comprises multiple dense layers with dropout regularization [76,77]. The input layer yields 128 units and 1536 parameters, succeeded by

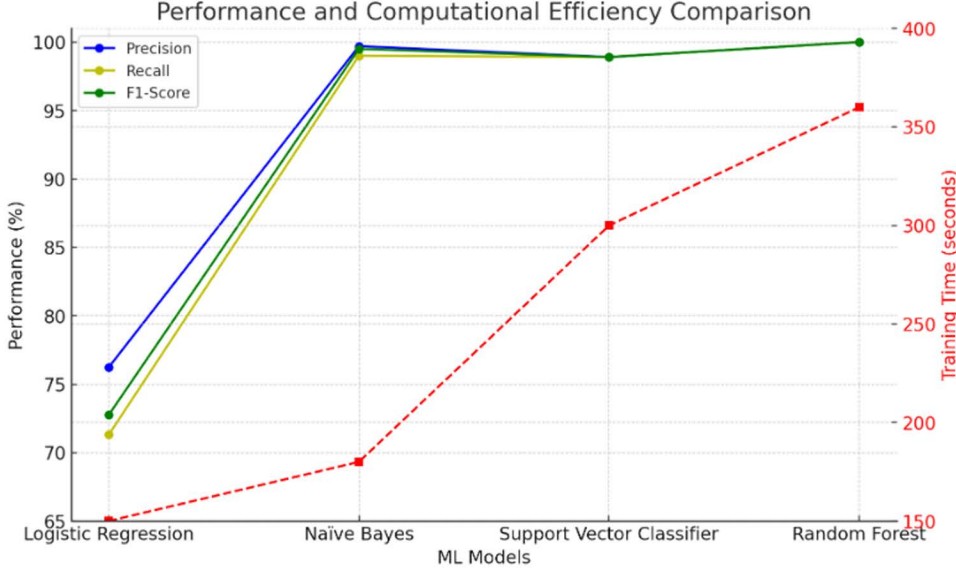

**Fig 16. Performance & computational efficiency comparison of ML models.**

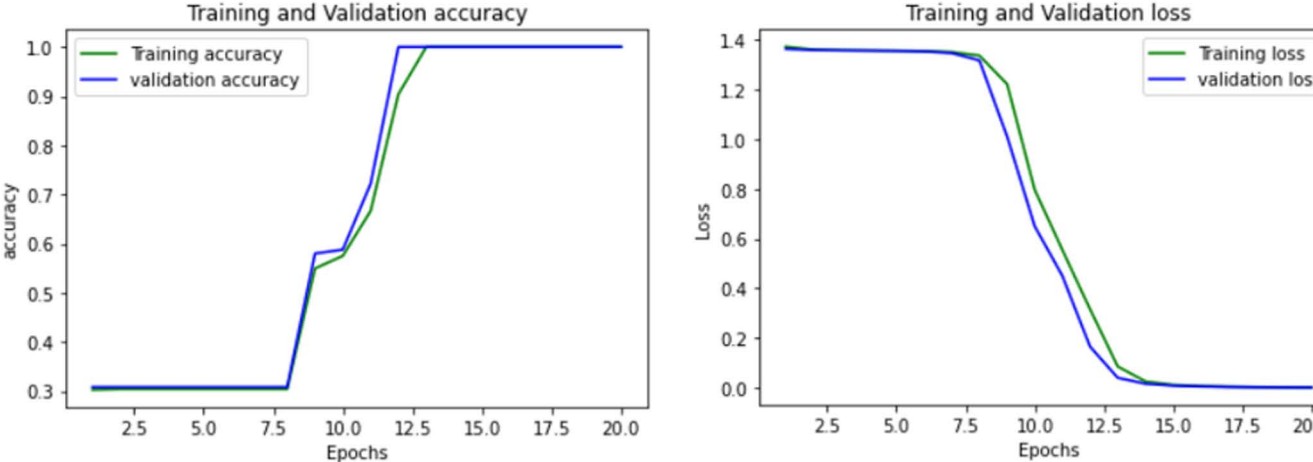

**Fig 17. Training -validation accuracy & lost for ANN.**

a 30% dropout layer. A subsequent dense layer generates 64 units with 8256 parameters, followed by another dropout layer. Another dense layer with 32 units and 2080 parameters is introduced, employing a Leaky ReLU activation function, dropout rate of 0.3 and L2 regularization 0.001. An additional dense layer with 16 units and 528 parameters is added, also utilizing Leaky ReLU activation and dropout. The final layer consists of 4 output units for classification, with 68 parameters and a SoftMax activation for multiclass output. The model encompasses a total of 12,468 trainable parameters and no non-trainable parameters. Training occurred over 20 epochs, attaining 100% accuracy and a minimal validation loss of 0.0024%, as depicted in Fig 17. Nevertheless, the training duration was significantly longer than conventional machine learning models, requiring approximately 920 seconds for completion.

## Conclusion

This research meticulously outlines the architecture and workflow of an ML/AI model, leveraging CPM data from a DAQ system. Through comprehensive Exploratory Data Analysis (EDA), Feature Engineering, and Data Visualization, we gained profound insights into the data. Key outcomes include insightful data visualization through univariate, bivariate, and multivariate analyses, which provided a deep understanding of the data. Robust data cleaning was ensured through feature engineering expertise, and an optimal train-test split of 70−30% was employed for model selection, followed by standardization of training and testing data. Hypothesis testing using Z-test and ANOVA confirmed the absence of bias post-preprocessing.

These steps ensured the data's readiness for training ML and Neural Network models. Notably, Neural Networks achieved up to 100% accuracy after 20 epochs, with a validation loss of just 0.0024% but at the cost of high computation time. Nonetheless, the Random Forest classifier exhibited a propensity to overfit, presenting difficulties for production deployment due to the complexity of the data and computation time. Among ML classifiers, the Naive Bayes Classifier outperformed the Support Vector Classifier and Logistic Regression Classifier based on accuracy and optimal computational time. In conclusion, the methodology of EDA, Feature Engineering, Standardization, and Hypothesis Testing has set a solid foundation for high-performing ML and AI algorithms. Deployment scalability is limited by integration challenges with existing DAQ systems. Additionally, inconsistencies in sensor compatibility and calibration may affect model performance across diverse industrial CPM. Future work will focus on training and testing these models to real time deployment. The reliability will be more on hardware's with higher Graphics Processing Unit (GPU) that support IIOT (Industrial Internet of Things) and also cloud computing infrastructure.

## Author contributions

**Conceptualization:** Gaurav Sandeep Dave.

**Data curation:** Gaurav Sandeep Dave.

**Formal analysis:** Gaurav Sandeep Dave, Amar Pradeep Pandhare, Dhananjay Vasant Khankal.

**Investigation:** Gaurav Sandeep Dave, Atul Prabhakar Kulkarni, Dhananjay Vasant Khankal.

**Methodology:** Gaurav Sandeep Dave, Amar Pradeep Pandhare.

**Supervision:** Amar Pradeep Pandhare, Atul Prabhakar Kulkarni, Dhananjay Vasant Khankal.

**Validation:** Gaurav Sandeep Dave, Dhananjay Vasant Khankal.

**Writing – original draft:** Gaurav Sandeep Dave.

**Writing – review & editing:** Gaurav Sandeep Dave.

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
