## [Decision Letter · Decision Letter 0]

Dear Dr. Dave,

We look forward to receiving your revised manuscript.

Kind regards,

John Adebisi, Ph.D

Academic Editor

PLOS ONE

Journal Requirements:

5. We note you have included a table to which you do not refer in the text of your manuscript. Please ensure that you refer to Table 4 in your text; if accepted, production will need this reference to link the reader to the Table.

Reviewers' comments:

Reviewer's Responses to Questions

**Comments to the Author**

1. Is the manuscript technically sound, and do the data support the conclusions?

Reviewer #1: Partly

Reviewer #2: Yes

2. Has the statistical analysis been performed appropriately and rigorously?

Reviewer #1: No

Reviewer #2: Yes

3. Have the authors made all data underlying the findings in their manuscript fully available?

Reviewer #1: No

Reviewer #2: Yes

4. Is the manuscript presented in an intelligible fashion and written in standard English?

Reviewer #1: No

Reviewer #2: Yes

Reviewer #1: The article presents a machine learning (ML) and deep learning (DL) framework for predictive maintenance of centrifugal pump machines (CPM), focusing on automating condition monitoring and fault detection. The authors highlight the limitations of traditional monitoring methods, such as manual data analysis, which is time-consuming and lacks adaptability across various operating conditions. By implementing a five-point summary of the data, exploratory data analysis (EDA), feature engineering, and data visualization, the study seeks to enhance data quality before model training. The ML models used, including random forest and deep learning classifiers, are evaluated based on metrics such as accuracy and F1-score, achieving high precision in detecting pump conditions. The proposed approach demonstrates an efficient, robust alternative for condition monitoring in industrial applications, with the potential for broad adaptation in predictive maintenance workflows.

Attached are my recommendations for the article.

Reviewer #2: (1) The introduction discusses general challenges in centrifugal pump optimization but can be improved by explicitly stating the unique contributions of this study. For instance, clearly highlight how the Dewsoft FFT DAQ system and the applied machine learning techniques improve upon existing methods in terms of predictive accuracy and operational efficiency.

(2) The section on hypothesis testing (page 14, Figure 5) is well-structured but could benefit from a brief comparison of the results obtained through Z-tests and ANOVA for different preprocessing scenarios. Explain how these tests validate the absence of bias and their implications for the robustness of the machine learning models.

(3) Some figures, such as the heatmap on page 20 (Figure 12), are informative but lack sufficient descriptions to guide readers unfamiliar with correlation matrices. Add a legend or a few explanatory sentences to clarify key observations, such as which correlations are critical for predictive maintenance.

(4) While the paper presents model performance (Tables 1–4, pages 24–29), the discussion should include a comparison of computational efficiency or training times across models like Naïve Bayes, SVC, and Random Forest. This would provide practical insights into model selection for real-time deployment.

(5) The manuscript effectively discusses data preprocessing and modeling but should connect these findings to practical scenarios. For example, explain how the insights from clustering (e.g., Figures 13 and 14 on page 22) could inform maintenance schedules or reduce downtime in industrial settings.

(6) I think the paper could benefit from the follwoing papers that greatly inestigated the optimal ML workflow:

https://doi.org/10.15530/urtec-2024-4044244

https://doi.org/10.1016/j.geothermics.2024.103028

https://doi.org/10.2118/219231-MS

https://doi.org/10.3390/en15238835

**Do you want your identity to be public for this peer review?** For information about this choice, including consent withdrawal, please see our Privacy Policy

Reviewer #1: No

Reviewer #2: **Yes: ** Taha Yehia

---

## [Author Response · Author response to Decision Letter 1]

15 Jan 2025

Dear Dr. Adebisi,

Thank you for your detailed feedback and for providing the opportunity to revise and resubmit my manuscript titled "Innovative Data Techniques for Centrifugal Pump Optimization with Machine Learning & AI Model" (PONE-D-24-46901). I have carefully reviewed the comments from the reviewers and the editorial team and have made the necessary revisions to address the points raised. Attached to this email, you will find the following documents as per your submission guidelines:

1. A rebuttal letter detailing my responses to each of the reviewers' comments and the changes made in the manuscript.

2. A marked-up copy of the revised manuscript with tracked changes.

3. A clean version of the revised manuscript without tracked changes.

I believe the revisions have significantly improved the manuscript, and I hope it now meets the publication criteria of PLOS ONE. Please let me know if there are any additional steps or further clarifications required.

Best regards,

Gaurav Dave

---

## [Decision Letter · Decision Letter 1]

Dear Dr. Dave,

We look forward to receiving your revised manuscript.

Kind regards,

John Adebisi, Ph.D

Academic Editor

PLOS ONE

**Comments from PLOS Editorial Office** : We note that one or more reviewers has recommended that you cite specific previously published works in the current and previous rounds of revision. As always, we recommend that you please review and evaluate the requested works to determine whether they are relevant and should be cited. It is not a requirement to cite these works and you may remove any added citations before the manuscript proceeds to publication. We appreciate your attention to this request.

**Additional Editor Comments:**

Dear Authors

Your manuscript has been given a major revision due to some observations by one of the reviewers especially who have requested a Major revision to improve clarity, methodological rigor, and language. The authors should address inconsistencies in statistical interpretations, ensure ML model transparency, and refine figures/tables for readability.

Reviewers' comments:

Reviewer's Responses to Questions

**Comments to the Author**

Reviewer #2: (No Response)

Reviewer #3: All comments have been addressed

Reviewer #4: (No Response)

2. Is the manuscript technically sound, and do the data support the conclusions?

Reviewer #2: Yes

Reviewer #3: Yes

Reviewer #4: Partly

3. Has the statistical analysis been performed appropriately and rigorously?

Reviewer #2: Yes

Reviewer #3: Yes

Reviewer #4: Yes

4. Have the authors made all data underlying the findings in their manuscript fully available?

Reviewer #2: (No Response)

Reviewer #3: Yes

Reviewer #4: Yes

5. Is the manuscript presented in an intelligible fashion and written in standard English?

Reviewer #2: (No Response)

Reviewer #3: Yes

Reviewer #4: No

Reviewer #2: (No Response)

Reviewer #3: Authors have addressed all the comments successfully. Good luck for future work in this research area

Reviewer #4: Major revision required to improve clarity, methodological rigor, and language. The authors should address inconsistencies in statistical interpretations, ensure ML model transparency, and refine figures/tables for readability.

• The abstract presents a broad summary but does not include numerical results. Sentences like "this approach improves operational efficiency" need quantification (e.g., percentage increase in efficiency, reduction in downtime).

• The phrase "the quality of data recorded plays the important role" is grammatically incorrect and should be revised for readability. Additionally, "machine learning (ML) & Deep Learning models (DL)" should be consistently formatted throughout the paper.

• Weak Problem Statement. The introduction mentions challenges in centrifugal pump monitoring but does not specify why current machine learning techniques are insufficient. The research problem should clearly define the existing gaps in literature with specific citations.

• Lack of Theoretical Background. The study mentions the application of ML and AI but does not provide a theoretical framework linking these methods to predictive maintenance or anomaly detection in pump systems. Referencing established theories in condition monitoring and fault detection would strengthen the argument.

• The same references appear multiple times without adding new insights. The literature review should be more structured, categorizing studies into specific themes (e.g., feature engineering, predictive maintenance, deep learning architectures).

• The paper mentions feature engineering techniques but does not explain why specific feature selection methods (e.g., PCA, mutual information) were chosen over others. A justification for the methodology is needed.

• Some papers are important to cite in the literature.

1. Active learning-based machine learning approach for enhancing environmental sustainability in green building energy consumption

2. Integrating machine and deep learning technologies in green buildings for enhanced energy efficiency and environmental sustainability

• Inadequate justification of data selection and sample size.

• The dataset consists of 70,062 rows and 13 columns, but the paper does not explain why this data size is appropriate for training ML models. Was a power analysis or cross-validation strategy applied to confirm adequacy?

• The study mentions data cleaning and transformation, but does not describe how missing values, outliers, or imbalanced classes were handled. These preprocessing steps are crucial for ML performance and should be explicitly detailed.

• While the paper presents means and standard deviations, there is no explanation of what these values imply in the context of pump optimization. Does higher variance in sensor data indicate potential faults?

• The regression analysis claims that sensor temperature has a strong correlation with performance, but this contradicts previous claims that flow rate is the primary predictor

• The study reports high ML accuracy (99.99%), but does not compare against simpler baseline models (e.g., logistic regression). Without a benchmark comparison, the claims about model superiority are unconvincing.

• The deep learning model achieved 100% accuracy on both training and testing data, which suggests severe overfitting. Did the authors apply regularization techniques (dropout, L2 penalty) to mitigate overfitting?

• The conclusion overstates the success of ML models without acknowledging limitations, such as potential biases in sensor data, generalizability to different pump types, and interpretability challenges.

• Significant grammatical errors and formatting inconsistencies.

**Do you want your identity to be public for this peer review?** For information about this choice, including consent withdrawal, please see our Privacy Policy

Reviewer #2: **Yes: ** Taha Yehia

Reviewer #3: No

Reviewer #4: No

---

## [Author Response · Author response to Decision Letter 2]

28 Apr 2025

Thank you for reviewing my manuscript and for giving me the opportunity to make the necessary updates. I wanted to let you know that I have carefully made all the required changes and uploaded the files as suggested by the reviewers.

I truly appreciate your guidance and support throughout this process. Looking forward to your feedback.

I truly appreciate your guidance and support throughout this process.

---

## [Decision Letter · Decision Letter 2]

Innovative Data Techniques for Centrifugal Pump Optimization with Machine Learning & AI Model.

PONE-D-24-46901R2

Dear Dr. Dave,

We’re pleased to inform you that your manuscript has been judged scientifically suitable for publication and will be formally accepted for publication once it meets all outstanding technical requirements.

Kind regards,

John Adebisi, Ph.D

Academic Editor

PLOS ONE

Additional Editor Comments (optional):

Reviewers' comments:

Reviewer's Responses to Questions

**Comments to the Author**

Reviewer #2: All comments have been addressed

Reviewer #3: All comments have been addressed

Reviewer #4: All comments have been addressed

2. Is the manuscript technically sound, and do the data support the conclusions?

Reviewer #2: Yes

Reviewer #3: Yes

Reviewer #4: Yes

3. Has the statistical analysis been performed appropriately and rigorously?

Reviewer #2: Yes

Reviewer #3: Yes

Reviewer #4: Yes

4. Have the authors made all data underlying the findings in their manuscript fully available?

Reviewer #2: Yes

Reviewer #3: Yes

Reviewer #4: Yes

5. Is the manuscript presented in an intelligible fashion and written in standard English?

Reviewer #2: Yes

Reviewer #3: Yes

Reviewer #4: Yes

Reviewer #2: (No Response)

Reviewer #3: Authors have addressed all the comments successfully. I wish him good luck for future work in area of AI.

Reviewer #4: After carefully reviewing the authors' responses and revisions, I find all concerns have been addressed thoroughly. The paper now meets the journal's standards, and I recommend acceptance in its current form.

**Do you want your identity to be public for this peer review?** For information about this choice, including consent withdrawal, please see our Privacy Policy

Reviewer #2: No

Reviewer #3: No

Reviewer #4: No

---

## [Editor Report · Acceptance letter]

PONE-D-24-46901R2

PLOS ONE

Dear Dr. Dave,

I'm pleased to inform you that your manuscript has been deemed suitable for publication in PLOS ONE. Congratulations! Your manuscript is now being handed over to our production team.

Kind regards,

on behalf of

Dr. John Adebisi

Academic Editor

PLOS ONE